# Decreased BAFF Receptor Expression and Unaltered B Cell Receptor Signaling in Circulating B Cells from Primary Sjögren’s Syndrome Patients at Diagnosis

**DOI:** 10.3390/ijms23095101

**Published:** 2022-05-04

**Authors:** Stefan F. H. Neys, Gwenny M. Verstappen, Hendrika Bootsma, Frans G. M. Kroese, Rudi W. Hendriks, Odilia B. J. Corneth

**Affiliations:** 1Department of Pulmonary Medicine, Erasmus MC University Medical Center, 3015 GD Rotterdam, The Netherlands; s.neys@erasmusmc.nl; 2Department of Rheumatology and Clinical Immunology, University of Groningen, University Medical Center Groningen, 9700 RB Groningen, The Netherlands; g.m.p.j.verstappen@umcg.nl (G.M.V.); h.bootsma@umcg.nl (H.B.); f.g.m.kroese@umcg.nl (F.G.M.K.)

**Keywords:** Bruton’s tyrosine kinase (BTK), B cell receptor (BCR) signaling, B-cell activating factor receptor (BAFFR, BR3), phosphoflow cytometry, primary Sjögren’s syndrome (pSS), non-Sjögren sicca

## Abstract

Animal models of autoimmunity and human genetic association studies indicate that the dysregulation of B-cell receptor (BCR) signaling is an important driver of autoimmunity. We previously showed that in circulating B cells from primary Sjögren’s syndrome (pSS) patients with high systemic disease activity, protein expression of the BCR signaling molecule Bruton’s tyrosine kinase (BTK) was increased and correlated with T-cell infiltration in the target organ. We hypothesized that these alterations could be driven by increased B-cell activating factor (BAFF) levels in pSS. Here, we investigated whether altered BCR signaling was already present at diagnosis and distinguished pSS from non-SS sicca patients. Using (phospho-)flow cytometry, we quantified the phosphorylation of BCR signaling molecules, and investigated BTK and BAFF receptor (BAFFR) expression in circulating B cell subsets in an inception cohort of non-SS sicca and pSS patients, as well as healthy controls (HCs). We found that both BTK protein levels and BCR signaling activity were comparable among groups. Interestingly, BAFFR expression was significantly downregulated in pSS, but not in non-SS sicca patients, compared with HCs, and correlated with pSS-associated alterations in B cell subsets. These data indicate reduced BAFFR expression as a possible sign of early B cell involvement and a diagnostic marker for pSS.

## 1. Introduction

B cells contribute to autoimmune diseases through several key effector functions, including autoantibody production, autoantigen presentation, and the secretion of pro-inflammatory cytokines. B cell receptor (BCR) signaling plays a central role in B cell survival and function [1]. Dysregulated BCR signaling may be a driver of autoimmunity, as B-cell specific overexpression of Bruton’s tyrosine kinase (BTK) induces a spontaneous autoimmune phenotype resembling systemic lupus erythematosus (SLE) and Sjögren’s syndrome (SS) in mice [2]. In humans, genes encoding BCR signaling proteins, such as *LYN*, *BLK*, *BANK1*, *PTPN22*, and *PXK*, have been identified as genetic risk factors for SLE and primary (p)SS [3,4,5,6,7].

We have recently shown that BTK protein levels are increased in circulating B cells from several autoimmune disease patients, including rheumatoid arthritis (RA), pSS, and glomerulonephritis with polyangiitis (GPA) [8,9]. These levels were associated with high serum autoantibody levels (in RA and pSS) and disease activity (in GPA). Furthermore, increased BTK protein expression correlated with increased proportions and activation status of circulating pathogenic T cells in RA, and with T-cell infiltration in the parotid glands from pSS patients [8]. Increased BTK protein also coincided with increased basal phosphorylation levels of BTK, and with an increased sensitivity for BCR stimulation, particularly in naïve B cells [8,9]. Together, these findings indicate a possible role for dysregulated BCR signaling in the evolution of B cell hyperactivity in pSS.

B cell hyperactivity in pSS patients is illustrated by clinical manifestations, such as hypergammaglobulinemia, and a higher risk of developing non-Hodgkin B cell lymphoma. In the salivary glands, one of the main target tissues of pSS, B cell hyperactivity is reflected by the formation of ectopic germinal centers, lymphoepithelial lesions and a plasma cell shift towards immunoglobulin (Ig)G [10,11]. How exactly B cell hyperactivity in pSS evolves remains unknown. In a previous cross-sectional study of BTK expression in pSS, we examined patients with high systemic disease activity, who were selected for two clinical trials. In approximately two thirds of these patients, we found increased BTK protein expression in total circulating B cells [8].

B-cell activating factor (BAFF) is considered an important factor in driving B cell hyperactivity in pSS, as levels are increased both systemically and locally in inflamed glands from pSS patients, and correlate with disease activity [12,13,14,15,16,17,18]. In addition, BAFF-transgenic mice spontaneously develop an SLE/SS-like phenotype [12,19]. Since the signaling pathways of the BAFF receptor (BAFFR), the most important receptor for BAFF, and the BCR positively regulate one another [20,21,22,23], we hypothesized that increased BAFF levels could be driving enhanced BTK levels and BCR signaling in pSS patients. In the present study, we aimed to investigate whether BAFFR and BTK protein expression, and BCR signaling activity are already altered at diagnosis, and can be used as a marker to distinguish pSS patients from non-SS sicca patients. The latter group represents patients with sicca (dryness) symptoms who do not fulfil classification criteria for pSS. We compared the expression of BTK protein and markers associated with B cell activation and survival, such as CD86, HLA-DR, and BAFFR, in circulating B cells between pSS patients at diagnosis, non-SS sicca patients, and healthy controls (HCs). Using phosphoflow cytometry we also compared BCR signaling between groups, both at basal level and upon stimulation. Finally, we correlated our findings to disease parameters, including the European Alliance of Associations for Rheumatology (EULAR) Sjögren Syndrome Disease Activity Index (ESSDAI) [24].

## 2. Results

### 2.1. BTK Levels and Basal BCR Signaling Activity Are Similar between Non-SS Sicca and pSS Patients

To study if an activated B cell phenotype, in terms of increased BTK and/or CD86 expression, was already present in recently diagnosed pSS patients, we analyzed peripheral blood mononuclear cells (PBMCs) from 27 pSS patients and compared these with 30 non-SS sicca patients (patient group 1; Table 1).

Intracellular expression of BTK and surface expression of the activation marker CD86 were determined in several B cell subsets using flow cytometry (for gating strategy: see Appendix A). We found increased numbers of transitional B cells and plasma blasts, and decreased proportions of the memory B cell subsets in pSS patients compared with non-SS sicca patients (Appendix A). However, the expression of BTK and CD86 was similar between non-SS sicca and pSS patients, both in naïve B cells and in all memory B cell subsets analyzed (Figure 1A,B; shown for total memory B cells). No correlation was observed between BTK and CD86 expression within all samples measured (Figure 1C), nor between BTK expression and the ESSDAI score in pSS patients (Figure 1D).

Next, we used phosphoflow cytometry to compare basal phosphorylation of BCR signaling molecules in total B cells from non-SS sicca and pSS patients (for gating strategy: see Appendix A). Phosphorylation of BTK (at tyrosine (Y)551), its downstream target phospholipase Cγ2 (PLCγ2; at Y759), and the BTK-upstream spleen tyrosine kinase (SYK; at Y348), was equal between circulating B cells from non-SS sicca and pSS patients (Figure 1E–G). We observed a positive correlation between BTK protein expression and BTK phosphorylation in B cells from all patients (non-SS sicca and pSS together; Figure 1H). However, no significant correlations were observed between the expression of BTK and basal phosphorylation levels of PLCγ2 and SYK in circulating B cells (Figure 1I,J).

Together, these findings indicate that both BTK protein expression and basal BCR signaling activity are comparable between recently diagnosed pSS and non-SS sicca patients.

### 2.2. BTK Levels and BCR Signaling Responsiveness Are Unaltered in Non-SS Sicca and pSS Patients Compared with HCs

Since we observed a difference in BTK expression level between pSS patients with high systemic disease activity and HCs in a previous study [8], the observed similarity in BTK expression and basal BCR signaling activity between non-SS sicca and pSS patients in the current study was rather unexpected. For this reason, we next investigated whether BTK expression and basal BCR signaling activity in B cells from non-SS sicca and pSS patients at diagnosis were different from HC B cells. Non-SS sicca and pSS patients from patient group 1 with various BTK expression levels (low to high) were randomly selected and we compared BCR signaling responsiveness with HCs (*n* = 10 for each group). The selected pSS group consisted of both patients with low (<5) and moderate-to-high (≥5) ESSDAI scores (Table 2).

Similar trends in B cell subset shifts, as observed in patient group 1 between pSS and non-SS sicca, were observed between pSS patients, non-SS sicca patients, and HCs (gating strategy in Appendix A), although differences were not statistically significant, most likely due to the smaller group size (Appendix A). Similar to the first patient group, no significant difference in BTK expression was observed between circulating B cells from non-SS sicca and pSS patients, and BTK levels in B cells from pSS and non-SS sicca patients were comparable to those in HCs (Figure 2A). Next, expression of the activation markers CD86 and HLA-DR was determined. Surface CD86 expression in naïve B cells was slightly reduced in pSS patients compared with HCs (Figure 2B). No differences were observed in surface HLA-DR expression between the three groups (Figure 2C). Similar to our reported findings in pSS patients with a high systemic disease activity [8], a weak positive correlation was observed between BTK and CD86 expression when combining all samples (shown for total B cells in Appendix A). No correlation was found between BTK and HLA-DR expression (shown for total B cells in Appendix A).

Next, we used phosphoflow cytometry to compare the basal BCR signaling activity in different B cell subsets between non-SS sicca patients, pSS patients, and HCs (gating strategy in Appendix A). A decrease in the phosphorylation of BTK at position Y551, which is thought to be mainly phosphorylated by SRC-like kinases and SYK [25], was observed in both non-SS sicca and pSS patients compared with HCs (Appendix A). However, no differences were observed in the basal phosphorylation of BTK at its autophosphorylation site Y223, nor of PLCγ2, or SYK (Appendix A).

To identify differences in responsiveness to BCR stimulation between groups, PBMCs were stimulated in vitro with anti-immunoglobulin (α-Ig) F(ab’)_2_ fragments, and phosphorylation of BTK (Y223), PLCγ2 (Y759), and SYK (Y348) was determined. In both unstimulated and stimulated conditions, phosphorylation of PLCγ2 was lower in circulating B cells from non-SS sicca and pSS patients, compared with HCs (Figure 3A). This same trend was observed for pBTK (at Y223; Figure 3B). No significant alterations were found in the phosphorylation of SYK (Figure 3C). As a measure of BCR responsiveness, the stimulation ratio ((gMFI from α-Ig-stimulated B cells)/(gMFI from unstimulated B cells)) was determined for each signaling molecule. Ultimately, this resulted in an equal responsiveness (as inferred from the stimulation ratio) of BTK and PLCγ2 between non-SS sicca patients, pSS patients, and HCs following BCR stimulation (Figure 3D,E). A small but significant increase in the responsiveness of SYK following BCR stimulation was observed in naïve B cells from non-SS sicca patients, compared with HCs (Figure 3F).

Together, these data indicate that BTK levels and BCR signaling responsiveness are essentially unaltered in recently diagnosed non-SS sicca and pSS patients compared with HCs.

### 2.3. BAFF Receptor Downregulation Is an Early pSS Marker

BAFF is considered an important factor in pSS pathogenesis and levels are increased in patients’ circulation [12,13,14,15,16]. Next to BCR signaling, BTK is also involved in the downstream signaling of the most important receptor for BAFF, the BAFFR [22]. Therefore, we studied the surface expression of BAFFR on several B cell subsets in non-SS sicca patients, pSS patients, and HCs. Interestingly, BAFFR surface expression was significantly decreased on both naïve and memory B cell populations from pSS patients compared with HCs (Figure 4A; only total memory B cells shown). BAFFR expression on B cells from non-SS sicca patients reached intermediate levels, whereby its expression on memory B cells was significantly different from pSS patients. Positive correlations were observed for BAFFR expression levels with the proportions of circulating IgD^+^ memory B cells and the proportions of plasmablasts in all samples measured (Figure 4B,C). No correlation was found between BTK and BAFFR expression (Figure 4D). Interestingly, a tendency towards a negative correlation between BAFFR expression and the ESSDAI was observed in pSS patients (Figure 4E). No correlation was found between the proportions of plasmablasts and the ESSDAI score (data not shown).

Together, these data indicate decreased BAFFR expression as a possible sign of early B cell involvement and diagnostic marker for pSS.

## 3. Discussion

We have previously shown in pSS patients with high systemic disease activity that BTK protein levels in circulating B cells were increased compared with HCs. To study whether this hyperactive B cell phenotype is already present in the early stages of the disease, we compared the B cell phenotype of pSS patients at the time of diagnosis with non-SS sicca patients and HCs. In contrast to our previous findings, these pSS patients showed BTK expression levels comparable to non-SS sicca patients and HCs, and no differences were observed in BCR signaling activity. However, we found that expression of BAFFR on circulating naïve and memory B cells was decreased in pSS patients, compared with non-SS sicca patients and HCs, and correlated with pSS-associated alterations in B cell subsets.

A defective tolerance leading to the survival and activation of autoreactive B cells, already present in the newly emerging transitional and naïve B cell stages, is thought to play a central role in the pathogenesis of systemic autoimmune diseases, including pSS [26,27]. Disturbances in lymphocyte counts and B cell subpopulations in pSS patients, as we have shown here, have been reported previously when compared with HCs [28,29,30,31,32,33]. Increasing percentages of transitional and naïve B cells, and decreased proportions of memory B cells in the circulation of pSS patients are thought to be the result of (autoreactive) memory B cells migrating and accumulating in the affected tissues (e.g., salivary glands) [28,34]. Similar alterations have been reported for non-SS sicca patients, indicating that possibly in a subgroup of non-SS sicca patients, who represent ‘incomplete’ pSS patients, these alterations are already present at an early stage of disease development [30,35]. However, possibly due to a lack of power, we did not observe such alterations when comparing non-SS sicca patients with HCs in our second patient group. Furthermore, the non-SS sicca group is a heterogeneous population of patients with subjective sicca symptoms who were referred to our Sjögren expertise center because of a clinical suspicion of pSS. Autoimmune features, such as a positive ANA titer or Raynaud’s phenomenon, were present in part of these patients, which could also be an explanation for the lack of difference between the non-SS sicca and pSS group.

Increasing evidence points towards a possible role for enhanced BTK activity in circulating B cells from systemic autoimmune patients [8,9,36]. We have previously shown that BTK levels in circulating B cells were elevated in a large proportion of pSS patients, recruited for phase II trials of abatacept or rituximab treatment [8]. Increased BTK levels were already present in the naïve B cell population and correlated with T cell infiltration in affected glands, and with total IgM and rheumatoid factor serum levels [8]. In the current study, we investigated whether aberrations in BCR signaling activity could be detected at diagnosis. Different from our findings in pSS patients with high systemic disease activity, BTK levels were not increased in B cells from recently diagnosed pSS patients, when compared with non-SS sicca patients or HCs. BCR signaling activity was also unaltered between patients and HCs. A major difference between the current and the initial study was the level of systemic disease activity (median ESSDAI of 4 vs. 11, respectively). In addition, pSS patients in the initial study had a longer disease duration, were all anti-SSA positive and exhibited stronger serological abnormalities. In line with our current results, a previous study also showed no difference in BCR signaling activity between a cohort of pSS patients with relatively low systemic disease activity and HCs [37]. Hence, these findings suggest that BTK levels, and possibly BCR signaling, are only enhanced in circulating B cells from a fraction of patients with a high systemic disease activity, and are not (yet) altered at time of diagnosis. A pro-inflammatory environment or certain T cell stimuli could drive this phenotype, as would be in concordance with our previous findings that increased BTK levels in pSS B cells normalize following abatacept treatment [8].

BAFF is an important activation and survival factor for B cells. Evidence suggests that the BAFFR utilizes part of the BCR signaling pathway, including SYK and BTK, for its downstream signaling [22,23]. In pSS, BAFF is considered an important pathogenic factor, as increased levels are found in serum and correlate with disease activity [12,13,14,15,38]. It is thought that local production in inflamed salivary glands stimulates local autoreactive B cell survival and activation. In addition, mice overexpressing BAFF spontaneously develop an SLE/SS-like phenotype, with sialoadenitis, destruction of the salivary glands and decreased saliva production at older age [12,19]. Paralleling our findings, previous studies have shown a decreased expression of the BAFFR on B cells from systemic autoimmune disease patients, including pSS, SLE, and systemic sclerosis, where it is hypothesized that BAFFR downregulation on the surface is the result of increased BAFF levels [39,40]. The downregulation of BAFFR expression observed in a subset of non-SS sicca patients might also be an early event in these patients who are possibly prone to pSS development and do not (yet) fulfil pSS criteria. In our study, the anti-BAFFR clone 11C1 was used, which is thought to compete with binding of BAFF, but only at supra-physiological concentrations, not reached in patients [39,41]. Therefore, the reduced detection of the receptor is thought to be the result of downregulation of surface expression and not because of occupancy of the receptors. This same feature was observed in a transgenic mouse model, where autoreactive B cells downregulated BAFFR expression in response to BAFF without changing mRNA expression [42]. In pSS patients, *BAFFR* mRNA expression in circulating B cells was also unaltered compared with HCs, even though decreased BAFFR surface expression was observed, indicating this downregulation is possibly regulated by post-transcriptional mechanisms [39]. Internalization or shedding of the receptor following ligand binding is another option which could lead to a decreased detection [43,44]. Positive correlations were observed for BAFFR expression levels with the percentage of circulating IgD^+^ memory B cells and the proportions of plasmablasts. In inflamed glandular tissue, the production of pro-inflammatory cytokines, including BAFF, could facilitate the chemoattraction of memory B cells and plasmablasts and support local (autoreactive) B cell survival and differentiation [45,46], leading to decreased proportions in the circulation and supporting local accumulation of these cells. Whether this reduced BAFFR expression is a compensating mechanism, or rather a pathogenic process should be a topic for further study. A better understanding of BAFFR expression regulation and its downstream signaling might increase our understanding of pSS pathogenesis and lead to new therapeutic approaches.

Interest in the targeting of BAFF and the BAFFR in pSS is increasing. An open-label clinical trial with belimumab (anti-BAFF) in pSS showed positive effects on ESSDAI in 60% of the patients [47]. In a smaller clinical trial, a single dose of ianalumab (anti-BAFFR) in pSS patients also showed positive effects on ESSDAI [48], and a larger and more recent phase IIb clinical trial with ianalumab again reported a dose-dependent amelioration of disease, as measured by decreases in ESSDAI score [49]. In addition to the BAFF-BAFFR axis, BTK has also been considered as a potential target for pSS treatment. Although multiple clinical trials with BTK inhibitors in pSS are still recruiting and results are awaited (NCT04035668; NCT04186871), the limited effectiveness of BTK inhibition in a recent study might be partly explained by our current findings of unaltered BTK levels and BCR signaling activity in circulating B cells from pSS patients at diagnosis [50,51]. Nevertheless, targeting BTK in a subpopulation of pSS patients might still be beneficial, perhaps only in a fraction of patients with high systemic disease activity. Our study was also restricted to circulating B cells, and BTK inhibition could still ameliorate B cell activation, including autoantibody secretion, at inflamed tissue sites such as the salivary glands.

In this study, we have shown that pSS patients at diagnosis present with unaltered BTK levels and BCR signaling activity compared with non-SS sicca patients and healthy controls. However, BAFF receptor downregulation on B cells from pSS patients was observed. Diagnosis of pSS remains difficult, in particular in patients who do not show pSS-associated autoantibodies (anti-SSA and anti-SSB) [52]. Therefore, more knowledge into pSS pathogenesis is needed to identify novel biomarkers and potential therapeutic targets. The observations from the current study indicate the BAFF–BAFFR axis as a potential therapeutic target or diagnostic marker and should be a subject for future studies.

## 4. Materials and Methods

### 4.1. Patient Characteristics

Consecutive patients, referred to the University Medical Center Groningen (UMCG) for suspicion of pSS, were included in an inception cohort (*n* = 98), as previously described [53]. The main inclusion criteria were age ≥ 18 years and sicca complaints. pSS patients were classified based on the fulfilment of 2016 ACR/EULAR criteria for pSS [54]. Non-SS sicca patients were patients who did not fulfil these criteria. For the current study, we included pSS patients with available PBMC samples (*n* = 27) and 30 age-matched non-SS sicca patients (Table 1). A second cohort consisted of 10 non-SS sicca patients, 10 pSS patients (both including patients from patient group 1), and 10 age- and sex-matched HCs (Table 2). Blood was collected in lithium heparin tubes (BD Biosciences, San Jose, CA, USA) following informed consent. PBMCs were isolated using Ficoll-Paque™ density gradient separation (GE Healthcare, Chicago, IL, USA) and were stored at −196 °C. Informed consent was obtained according to the Declaration of Helsinki and the study was approved by the Medical Research Ethics Committees from the UMCG (METc2013.066) and the Erasmus MC University Medical Center Rotterdam (MEC-2021-0656).

### 4.2. Flow Cytometric Analysis

Cryopreserved vials with PBMCs were thawed, resuspended in ice-cold RPMI (Gibco, Thermo Fisher Scientific, Waltham, MA, USA) containing 5% FCS (Gibco; medium), and put into 96-well round-bottom plates.

#### 4.2.1. B Cell Surface Marker and Intracellular Staining

Next, 5.0 × 10^5^ cells were plated and washed with 0.5% BSA, 2 mM EDTA in PBS (MACS buffer) and were incubated with TruStain FcX (Biolegend, San Diego, CA, USA) to prevent binding of antibodies to Fc receptors. Cells were washed and subsequent staining for surface markers was performed. Cells were then stained with Fixable Viability Dye (Invitrogen, Waltham, MA, USA) and streptavidin, and 2% paraformaldehyde was used for fixation. Following permeabilization with 0.5% saponin (Sigma, St. Louis, MO, USA), BTK was stained intracellularly. A complete list of antibodies used is provided in Appendix A. Samples were measured on an LSR II flow cytometer (BD Biosciences).

#### 4.2.2. BCR Signaling Measurement by Phosphoflow Cytometry

Intracellular phosphoflow cytometry (phosphoflow) was used to determine the phosphorylation of intracellular signaling molecules, as described extensively elsewhere [55,56]. In short, 3.0–5.0 × 10^5^ cells were plated in medium. For the detection of basal phosphorylated (p)BTK-Y223 and -Y551, pSYK-y348, and pPLCγ2-Y759 levels, cells were incubated for 20 min on ice with a live/dead marker (Invitrogen), fixed for 10 min with the eBioscience^TM^ Foxp3/Transcription Factor Staining Fixation/Permeabilization Buffer (Invitrogen) and stained for extracellular and intracellular makers. For stimulation, cells were brought to 37 °C and 10 min before fixation, Fixable Viability Dye (Invitrogen) was added as a live/dead marker. Cells were incubated with 20 µg/mL goat F(ab’)_2_ anti-human immunoglobulin (α-Ig) fragments (Southern Biotech, Birmingham, AL, USA) in medium for 2 (pSYK) or 5 min (pBTK and pPLCγ2), or were left unstimulated as control. Fixation with the aforementioned Fixation/Permeabilization Buffer for 10 min at 37 °C terminated the stimulation and permeabilized the cells. Intracellular staining for 30 min at 4 °C in Permeabilization Buffer (Invitrogen) was then performed to stain for B cell markers. pSYK Y348-PE, pBTK Y551-PE, pBTK Y223-AF647, and pPLCγ2 Y759-AF647 were stained in Permeabilization Buffer at RT. Cells were resuspended in MACS buffer and were measured on an LSR II flow cytometer (BD Biosciences).

### 4.3. Statistical and Computational Analysis

Flow cytometric data were analyzed with FlowJo version 10.8.1 (BD Biosciences) and low-quality samples were removed from the comparison before any analysis. The geometric mean fluorescent intensity (gMFI) was used for quantification of the signal. To test for parametric distribution of data with each comparison, a Shapiro–Wilk test was performed. A one-way ANOVA with Tukey’s test for multiple comparisons or an unpaired t-test was used in the case of parametric distribution of the data. In the case of non-Gaussian distribution, a Kruskal–Wallis with Dunn’s test or a Mann–Whitney U was performed. Pearson or Spearman’s rank correlation coefficients were calculated for correlation analyses. *p*-values < 0.05 were considered statistically significant. GraphPad Prism 9 software (GraphPad Prism Inc., San Diego, CA, USA) was used for these analyses.

## Figures and Tables

**Figure 1 ijms-23-05101-f001:**
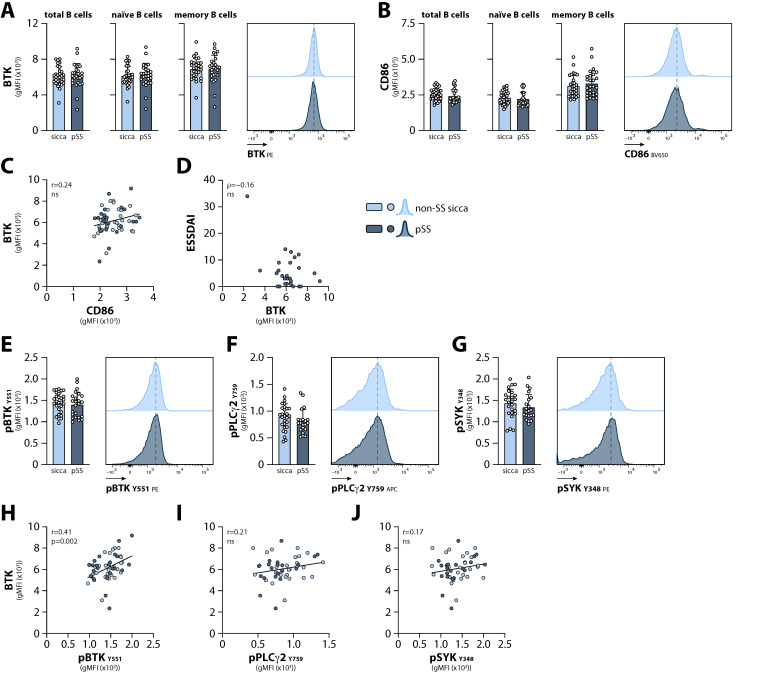
B cell receptor signaling activity in circulating B cells from non-SS sicca and pSS patients. Expression of (**A**) intracellular BTK protein and (**B**) surface CD86 in total, naïve, and memory B cells. Representative histograms are shown for total B cells. Bars indicate mean + SD. (**C**) Pearson correlation analysis between intracellular BTK protein expression and surface CD86 expression in total B cells from non-SS sicca and pSS patients. (**D**) Spearman’s rank correlation analysis between intracellular BTK protein expression in total B cells and the European Alliance of Associations for Rheumatology (EULAR) Sjögren Syndrome Disease Activity Index (ESSDAI) in pSS patients. (**E**–**G**) Basal phosphorylation of BTK (Y551), PLCγ2 (Y759), and SYK (Y348) in total B cells from non-SS sicca and pSS patients. Representative histograms are shown. Bars indicate mean + SD. (**H**–**J**) Pearson correlation analysis between intracellular BTK protein expression and (**H**) pBTK Y551, (**I**) pPLCγ2 (Y759), and (**J**) pSYK (Y348) in total B cells from non-SS sicca and pSS patients.

**Figure 2 ijms-23-05101-f002:**
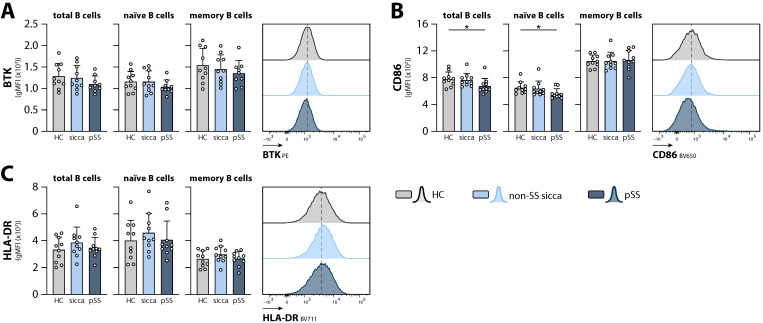
Expression of BTK and activation markers in circulating B cells from non-SS sicca patients, pSS patients, and HCs. Expression of (**A**) intracellular BTK protein, (**B**) surface CD86, and (**C**) surface HLA-DR on circulating total, naïve, and memory B cells. Representative histograms are shown for total B cells. Bars indicate mean + SD. * *p* < 0.05 by Kruskal–Wallis test with Dunn’s correction for multiple comparisons.

**Figure 3 ijms-23-05101-f003:**
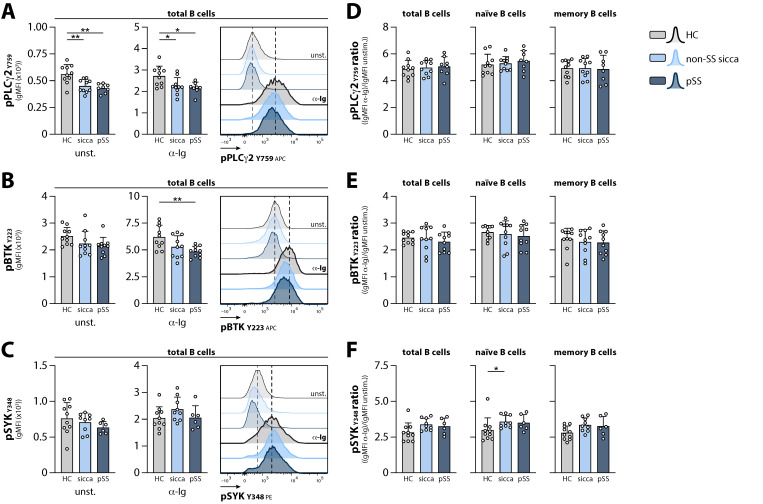
B cell receptor signaling responsiveness in circulating B cell subsets from non-SS sicca patients, pSS patients, and HCs. (**A**–**C**) Phosphorylation of (**A**) PLCγ2 (Y759), (**B**) BTK (Y223), and (**C**) SYK (Y348) in unstimulated (unst.) and α-Ig stimulated circulating total B cells. Representative histograms are shown. (**D**–**F**) The stimulation ratio for (**D**) pPLCγ2 (Y759), (**E)** pBTK (Y223), and (**F**) pSYK (Y348), as determined by dividing the phosphorylation signal from the α-Ig stimulated condition by the phosphorylation signal at the unstimulated condition. Data shown for total, naïve, and memory B cells. Bars indicate mean + SD. ** *p* < 0.01 and * *p* < 0.05 by Kruskal–Wallis test with Dunn’s correction for multiple comparisons.

**Figure 4 ijms-23-05101-f004:**
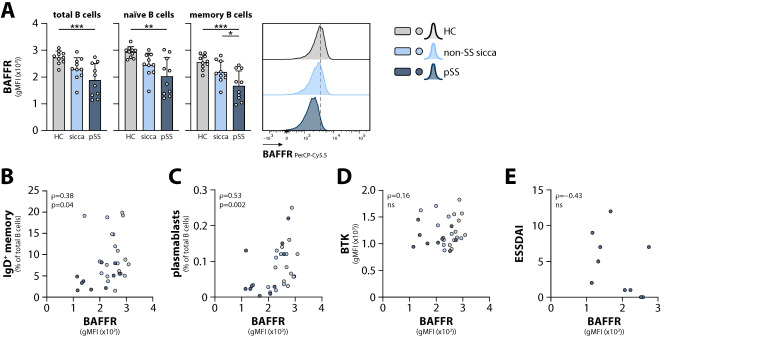
BAFFR expression on circulating B cells from non-SS sicca patients, pSS patients, and HCs. (**A**) BAFFR surface expression on total, naïve, and memory B cells. Representative histograms are shown for total B cells. Bars indicate mean + SD. (**B**–**E**) Spearman’s rank correlation analysis between surface BAFFR expression on total B cells and (**B**) proportion of IgD^+^ memory B cells, (**C**) proportion of plasmablasts, (**D**) intracellular BTK levels in total B cells from non-SS sicca patients, pSS patients and HCs, and (**E**) ESSDAI scores from pSS patients. *** *p* < 0.001, ** *p* < 0.01, * *p* < 0.05 by Kruskal-Wallis test with Dunn’s correction for multiple comparisons.

**Table 1 ijms-23-05101-t001:** Characteristics of patient group 1.

	non-SS Sicca	pSS
**N (female)**	30 (25)	27 (25)
**Age (years)**	50 (20–72)	54 (30–74)
**ESSDAI**	-	4 (0–34)
**Lymphocyte count (×10^3^/mL)**	2.03 (1.01–3.47)	1.3 (0.71–3.35)
**IgG (g/L)**	9.8 (7.6–16.6)	16.0 (9.7–32.3)
**RF positive**	1/30	14/26 *
**Anti-Ro/SSA positive**	1/30	20/27
**Anti-La /SSB positive**	0/30	11/27
**Biopsy positive ****	2/30	22/27

Values are given as median (range). RF: Rheumatoid factor. * Not determined for 1 patient ** Focus score ≥ 1 in minor or major salivary gland biopsy.

**Table 2 ijms-23-05101-t002:** Characteristics of patient group 2.

	HC	non-SS sicca	pSS
**N (female)**	10 (7)	10 (8)	10 (10)
**Age (years)**	50 (27–69)	48 (26–71)	48 (32–71)
**ESSDAI**	-	-	4 (0–12)
**Lymphocyte count (×10^3^/mL)**	-	2.68 (1.01–3.03)	1.85 (1.05–3.35)
**IgG (g/L)**	-	11.0 (6.7–14.9)	12.7 (9.7–26.7)
**RF positive**	-	1/10	4/10
**Anti-Ro/SSA positive**	-	0/10	6/10
**Anti-La/SSB positive**	-	0/10	3/10
**Biopsy positive ***	-	1/10	8/10

Values are given as median (range). RF: Rheumatoid factor. * Focus score ≥ 1 in minor or major salivary gland biopsy.

## Data Availability

The data presented in this study are available on request from the corresponding authors. The data are not publicly available due to privacy and ethical issues.

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
