# Peer review of "Decreased BAFF Receptor Expression and Unaltered B Cell Receptor Signaling in Circulating B Cells from Primary Sjögren’s Syndrome Patients at Diagnosis"

_ijms, 2022, doi:10.3390/ijms23095101_

Round 1

Reviewer 1 Report

In this study Stefan Neys et al. compared the activation status of B cells from patients with primary Sjögren’s syndrome (pSS) and non-SS sicca at the time of diagnosis (early disease stage). The study was presented as a follow-up investigation of previous work that had established increased BTK protein levels in circulating B cells of pSS patients with high systemic disease activity. Contrary to advanced pSS, authors did not observe any differences in BTK protein level and phosphorylation status, surface expression of the activation marker CD86, or BCR signaling in circulating B cells of patients with early onset pSS or non-SS sicca compared to healthy controls. However, authors identified a significant decrease in surface BAFFR expression in pSS B cells, which could qualify as early diagnostic marker for pSS. The authors discuss further that BAFFR downregulation might be a direct response to elevated BAFF levels, which are commonly increased during pSS and correlate well with diseases severity due to promoting the activation and survival of autoreactive B-cells.

The introduction and discussion are clearly explained and the conclusions are supported by the results. The identification of BAFFR expression as potential diagnostic marker for early stage pSS is an interesting finding. The work could be further strengthened if authors could correlate the decrease in BAFFR expression with elevated BAFF levels at time of diagnosis to confirm a direct relationship. Future studies could focus on a longitudinal study to determine whether early BAFFR downregulation could serve as prognostic marker to predict severity of pSS at later stages of disease.

The following section lists a few minor points, which would improve and complete the content. I would recommend accepting the article after minor revision.

Could authors comment more on the particular study design, in which they selected cohort 2 as a subset of cohort 1 for all subsequent analyses. In that way the first result section (with cohort 1) appears redundant with the subsequent results obtained with Cohort 2. Why did authors not decide to include more healthy controls and obtain stronger statistics for their comparisons with the full cohort 1?

Also, splitting up the cohorts in such a way leads to some statistical inconsistencies throughout the paper. For example, in Fig. 1B authors show no statistically significant difference in CD86 gMFI between total and naïve B-cells of pSS and non-SS sicca patients. However, with including just a smaller subfraction of patients in cohort 2 it appears now that there is a significant difference (Fig. 2B). Are these inconsistencies not just due to the smaller cohort size (n=10 instead of 27)?

How would authors comment on the relationship between plasmablast numbers and BAFFR expression, which was not entirely clear to me from the way the paper is written. The authors established that pSS patients exhibit a higher frequency of plasmablasts (Fig. S1D) and that BAFFR levels are decreased (Fig. 4A) However, in Fig. 4C BAFFR expression levels correlate positively with the proportion of plasmablasts. Intuitively, I would have expected an inverse correlation, in which decreased BAFFR levels would correlate with an increased frequency of plasmablasts. Maybe authors could comment in the text on the relationship of plasmablasts with pSS severity/ESSDAI score, if known.

Specific comments:

  • Figure legends do not state the figure number (however numbers are called out in main text)
  • The second section of the results part “ BTK levels and BCR signaling responsiveness are unaltered...” starts with a formatting issue – the font text is too small and starts with the word Scheme 8.
  • Typo in Materials and Methods Section 4.2. B cell surface marker and intracellular staining; last line: “Table SSamples”

Other than the above-mentioned points, the science seems sound and even though the authors present a substantial amount of ‘negative’ data they ultimately identified BAFFR expression as potentially new diagnostic marker for early stage pSS, which makes the results important and signal the need for further investigation.

Reviewer 2 Report

In this study, Neys et.al. reported that in contrast to their previous findings pSS patients showed comparable BTK levels and BCR signaling activity to healthy controls that might be explained by the lower disease activity of the patients at the time of diagnosis. The key novel finding is that BAFFR expression on circulating naïve and memory B cells was decreased in pSS patients compared to non-SS sicca patients and healthy controls. Moreover, a correlation was also observed between BAFFR expression and pSS-associated alterations in B cell subsets.

Generally, the data are interesting and well organized, the figures are easy to follow, the gating strategy is especially clearly shown, the patient number is adequate and the interpretation of the results is proper. 

I have only a few minor concerns:

  • In the figure legend the number of figures are missing.
  • The first sentence in the second part of the Results section (page 4) starts with “Scheme 8.” It must be a mistake, thus should be removed.
  • Is it possible to show a regression line in Fig. S2J such as as in Fig1.H-J? Or is not the sample size large enough to do so? I have the same question regarding Fig.4B and C.
  • Instead being a separate supplementary figure Fig.S4. should appear as a part of figure 4. There is enough place to show the data there and it would be easier to follow than searching for the figure in the supplementary material.
